# Novel anthropometric indices and cognitive function among iranian older adults: Bushehr Elderly Health (BEH) program

Shervin Mossavarali[1☯], Shahrzad Mohseni[2☯], Noushin Fahimfar[1,3], Negar Asaad Sajadi[4], Mahnaz Pejman Sani[2,5‡*], Farshad Sharifi[6], Mohammadreza Mohajeri-Tehrani[2], Bagher Larijani[2], Kazem Khalagi[1,7‡*], Iraj Nabipour[8]

1 Osteoporosis Research Center, Endocrinology and Metabolism Clinical Sciences Institute, Tehran University of Medical Sciences, Tehran, Iran, 2 Endocrinolog and Metabolism Research Center, Endocrinology and Metabolism Clinical Sciences Institute, Tehran University of Medical Sciences, Tehran, Iran, 3 Department of Epidemiology and Biostatistics, School of Public Health, Tehran University of Medical Sciences, Tehran, Iran, 4 Department of Biostatistics, School of Allied Medical Sciences, Shahid Beheshti University of Medical Sciences, Tehran, Iran, 5 Department of Endocrinology, Shariati Hospital, School of Medicine, Tehran University of Medical Sciences, Tehran, Iran, 6 Elderly Health Research Center, Endocrinology and Metabolism Population Sciences Institute, Tehran University of Medical Sciences, Tehran, Iran, 7 Obesity and Eating Habits Research Center, Endocrinology and Metabolism Clinical Sciences Institute, Tehran University of Medical Sciences, Tehran, Iran, 8 The Persian Gulf Marine Biotechnology Research Center, The Persian Gulf Biomedical Sciences Research Institute, Bushehr University of Medical Sciences, Bushehr, Iran

☯ These authors contributed equally to this work.
‡ These authors contributed as correspondence authors.
* mpsani@sina.tums.ac.ir (MPS), kkhalagi@yahoo.com (KK)

## Abstract

### Background

This cross-sectional study investigated novel anthropometric indices including the triglyceride-glucose (TyG) index, visceral adiposity index (VAI), and lipid accumulation product (LAP), which have been proposed as simple markers of insulin resistance. A factor contributing to cognitive impairment through mechanisms such as endothelial dysfunction, inflammation, and neuronal injury. Due to these Insulin resistance-related pathways, such indices may help identify older adults at higher risk of cognitive impairment. This study aimed to investigate the associations between TyG, VAI, and LAP and cognitive function in Iranian older adults using data from the Bushehr Elderly Health (BEH) program.

### Methods

This cross-sectional analysis used data from 2,426 participants aged ≥60 in the Bushehr Elderly Health program. Cognitive function was assessed using the functional assessment staging test (FAST), Mini-Cog, and category fluency test (CFT) questionnaires. Anthropometric indices (TyG, VAI, LAP) were calculated from clinical and laboratory measurements. Logistic regression, stratified by sex a priori

**Data availability statement:** The minimal data set for this study cannot be shared publicly due to ethical restrictions on sharing a de-identified data set, which contains potentially sensitive participant information. Data are available from the Osteoporosis Research Center of Tehran University of Medical Sciences (contact via emri-osteoporosis@sina.tums.ac.ir) for researchers who meet the criteria for access to confidential data. Access may be granted upon the submission of a research proposal and a valid ethics code to the Center.

**Funding:** The author(s) received no specific funding for this work.

**Competing interests:** The authors have declared that no competing interests exist.

and adjusted for demographic, lifestyle, and clinical confounders selected through backward stepwise regression, was used to examine associations with cognitive impairment.

## Results

Among participants (mean age: 65.34 ± 6.40 years, 52% female), cognitive impairment was identified in 63.9%. In males, higher VAI was associated with impairment in the FAST test [Odds ratio (OR): 1.601, 95% CI: 1.164–2.203; P = 0.004)], while in females, VAI in the second quartile was linked to lower odds of impairment (OR = 0.583, 95% CI: 0.341–0.997, p = 0.049). No consistent associations were observed for TyG or LAP.

## Conclusion

Although some sex- and test-specific associations were observed, particularly with VAI, these findings highlight the complex and multifactorial association between metabolic health and cognitive function. This implies that factors beyond insulin resistance may contribute. Further longitudinal research is needed to clarify these relationships.

## Introduction

Cognitive impairment is a disorder often associated with aging, characterized by difficulties in thinking, language, judgment, memory, and executive functions. These conditions can influence an individual's ability to perform daily activities [1]. With the global population aging, cognitive impairment has become a major public health concern, with about 20% of elderly individuals estimated to have cognitive impairment globally [2]. In Iran, recent studies report cognitive impairment in 15% to 60% of older adults, with some cases reaching up to 70% [3–6]. Although some therapies may slow its progression, there is currently no effective cure for dementia [7]. Consequently, developing reliable and accurate screening tools for the early detection of cognitive impairment remains crucial.

Several factors have been identified as potential risks for cognitive impairment in the elderly, including age, dyslipidemia, smoking, and chronic conditions such as diabetes and cerebrovascular disease [8]. Among these, insulin resistance (IR) plays a crucial role in many of these conditions [9,10], contributing to cognitive impairment through mechanisms such as blood-brain barrier dysfunction, endothelial dysfunction, inflammation, and neuronal damage [11]. Research suggests there is a bidirectional relationship between IR and cognitive impairment, highlighting the importance of closely monitoring both conditions [12].

Several biomarkers and indices have been proposed for assessing insulin resistance (IR), as traditional methods like the hyperinsulinemic-euglycemic clamp (HEC) and the homeostasis model assessment of insulin resistance (HOMA-IR) are costly and impractical for routine use [13,14], which highlights the need for practical

alternatives. Recently, novel anthropometric indices have been introduced as simple, affordable, and accessible options. The triglyceride-glucose (TyG) index has demonstrated greater accuracy in predicting IR and related conditions [15]. The lipid accumulation product (LAP), based on triglycerides and waist circumference, predicts cardiovascular risk, IR, metabolic syndrome, and diabetes more effectively than traditional measures [16,17]. Moreover, the visceral adiposity index (VAI) offers a precise assessment of visceral fat and is independently associated with IR and impaired insulin secretion [18,19].

Although numerous studies have explored the association between these novel anthropometric indices (TyG, VAI, and LAP) and cognitive function, few have examined all three indices together [20–22]. Due to the variation and high prevalence of cognitive impairment among the Iranian older adults, identifying the association between novel anthropometric indices and cognitive function is crucial for developing prevention and therapeutic strategies. Therefore, this study aimed to investigate the associations between TyG, VAI, and LAP and cognitive function among the Iranian older adults, using data from the Bushehr Elderly Health (BEH) program.

## Methods

### Study design

In this cross-sectional analysis, we used data from the second wave of baseline data collection (Stage 2) of the Bushehr Elderly Health (BEH) program. The BEH program is a prospective, multi-phase, population-based cohort study focused on investigating the non-communicable diseases (NCDs) among older adults residing in Bushehr, a southern province of Iran. In the first recruitment of the BEH program, 3,297 individuals aged 60 or more and living in 75 strata in Bushehr city were selected through a stratified cluster random sampling method and invited to participate. Eligibility criteria in this recruitment of the BEH program included being aged 60 or older, residency in Bushehr for at least one year prior to recruitment, no plans to relocate from Bushehr during the two years of follow-up, and sufficient physical and mental capacity to participate in the study.

After excluding non-resident participants and those who declined to take part, 3,000 individuals were enrolled in the first stage of the cohort. The second wave (Stage 2) of baseline measurements concentrated on cognitive disorders and musculoskeletal diseases. After accounting for participants who were lost to follow-up or had passed away, 2,426 individuals from the original cohort completed Stage 2 assessments. Briefly, the study protocol was approved by the relevant institutional ethics committee, and written informed consent had been obtained from all participants at enrollment. The detailed protocols of both stages have been published elsewhere [23,24].

Researchers obtained access to the Stage 2 dataset in May 2025, at which time the analysis for the present study was initiated. Only de-identified data were provided; therefore, the authors did not have access to any personal or identifying information during the analysis. The listwise deletion method was used to handle missing data. missing data were observed for several independent variables: education, socioeconomic index, and current smoking status (2 cases), blood pressure–lowering medication use (1,347 cases), and glucose-lowering medication use (713 cases). Importantly, there were no missing data for the primary variables of interest, lipid profile and cognitive impairment status.

### Data collection and variable definitions

Data on age, sex, income, and educational level were collected through interviews with direct survey questions conducted by trained healthcare workers. Level of education was categorized into five groups: illiterate, Primary school, high school, diploma, college degree or above.

The socio-economic index was constructed using PCA on employment status, insurance coverage, education level, household assets, income, and residence conditions. The first principal component explained 52.0% of the variance. Scores were grouped into quintiles and classified as very good, good, moderate, poor, or very poor. Further data

regarding lifestyle factors, medical history, medication use, and mental functional health were collected through comprehensive standardized questionnaires. Current smokers were defined as daily users of at least one cigarette, hookah, or pipe. Physical activity on an average weekday, including sports, work, and leisure, was assessed using a validated questionnaire based on metabolic equivalents (METs), and this instrument has been previously validated in older adults and in persian version [24–26]. Total daily METs were calculated by multiplying activity time by its MET level, classifying them as sedentary (1–1.39 METs), low active (1.4–1.59 METs), active (1.6–1.89 METs), or highly active (1.9–2 METs) [27].

## Cognitive tests

Cognitive function was measured with the functional assessment staging test (FAST), Mini-Cog test, and Category fluency test (CFT), which were translated and validated previously [28–30]. Cognitive impairment was defined as impairment in one or more of the specified tests.

The FAST scale measures seven levels of functioning, ranging from normal adult abilities to severe cognitive decline, with levels 6 and 7 further divided into substages, resulting in a total of 11 stages [31]. It evaluates functional decline primarily in dementia, focusing on the loss of ability to perform daily activities independently as dementia progresses. A FAST score of ≥3 indicated mild cognitive impairment with noticeable deficits in complex tasks [31]. In the Persian validation study, the instrument showed strong concurrent validity (Spearman's r = –0.879) and excellent reliability, demonstrated by a test–retest ICC of 0.946 and an inter-rater ICC of 0.997 [32].

The Mini-Cog, a short cognitive screening test, includes two tasks: a three-word recall test to assess memory and a clock-drawing test to evaluate cognitive functions such as visual-motor skills, language, and executive function. Cognitive impairment was indicated under the following conditions: (1) no words are recalled correctly, or (2) only one or two words are recalled and the clock-drawing test is scored as abnormal [33]. This test showed acceptable performance in the Persian validation study, demonstrating substantial inter-rater reliability (κ = 0.76), positive concurrent validity, and sensitivity and specificity of 88% and 62.8%, respectively [34].

The CFT assesses semantic verbal fluency by asking individuals to name as many animals as possible within 60 seconds. Cognitive impairment was defined according to education-adjusted cut-offs as follows: naming fewer than 9 animals for individuals with no formal education, fewer than 12 animals for those with some education, fewer than 13 animals for those with ≤12 years of education, and fewer than 16 animals for individuals with more than 12 years of education [35]. In this preliminary Persian validation study, the test demonstrated good discriminative validity for separating normal, MCI, and AD groups [36].

## Physical measurements

All measurements were performed by a trained nurse. Weight and height were measured following a standard procedure using a fixed stadiometer, with participants standing straight, facing forward, wearing light clothing, and without shoes. Body mass index (BMI) was measured by dividing weight (in kilograms) by height squared (in meters). Blood pressure (BP) was measured on the right arm using a standardized mercury sphygmomanometer after 15 min of seated rest. Systolic and diastolic blood pressure (SBP and DBP) were determined by recording the first and fifth Korotkoff sounds, respectively. The average of two readings was used to determine the participant's final BP. Waist circumference (WC) was measured at a point midway between the lowest rib and iliac crest in the standing position using a flexible tape.

## Laboratory measurements

Venous blood samples were taken by a qualified nurse after an 8–12 hours of fasting. Laboratory assessments, including fasting blood glucose (FBS), lipid profiles [(high-density lipoprotein cholesterol (HDL-C), low-density lipoprotein cholesterol (LDL-C), total cholesterol, and triglyceride (TG)], nd vitamin D were measured by standard protocols. Lipid

profile and FBS tests were performed using an enzymatic colorimetric method with a Pars Azmun kit (Pars Azmun, Karaj, Iran). Instruments were calibrated with daily internal quality control and periodic external quality assurance, with the second stage of data collection in October 2015 [24]. Dyslipidemia was defined as lipid-lowering medication use or any abnormal lipid value: total cholesterol ≥200 mg/dl, TG > 150 mg/dl, LDL-C > 130 mg/dl, or HDL-C < 40 mg/dl (women)/ < 50 mg/dl (men).

### Novel anthropometric indices

Anthropometric indices were calculated as follows, with standard conversion factors applied where relevant (1 mmol/L = 88.57 mg/dL for triglycerides; 1 mmol/L = 38.67 mg/dL for HDL-C):

LAP:

- For males: LAP = [WC (cm) − 65] × TG (mmol/L)

- For females: LAP = [WC (cm) − 58] × TG (mmol/L)

VAI:

- For males: VAI = [WC (cm)/ 39.68 + (1.88 × BMI (kg/m²))] × (TG (mmol/L)/ 1.03) × (1.31/ HDL-C (mmol/L))

- For females: VAI = [WC (cm)/ 36.85 + (1.89 × BMI (kg/m²))] × (TG (mmol/L)/ 0.81) × (1.52/ HDL-C (mmol/L))

TyG:

- TyG = Ln [TG (mg/dL) × FBS (mg/dL)/ 2] or LnTG(mmol/L) × FBS (mmol/L)/0.001253

### Statistical analysis

Statistical comparisons were performed with version 27 SPSS and version 14 STATA. Continuous normally distributed variables were compared with an independent samples T-Test. Comparison between categorical variables was done by the $\chi^2$ test. P-values of <0.05 were considered statistically significant. PCA analysis was performed to construct individuals' socioeconomic scores based on their socioeconomic variables, and then this score was divided into 5 quantiles. Logistic regression analyses were applied to compute the odds ratio (OR) and 95% CI to detect any potential association between the TyG, LAP, and VAI indexes (as continuous variables or categorized based on quartiles) and cognitive impairment. No covariates were adjusted for in the crude model. Backward stepwise method applied to identify important covariates in the adjusted model. Variables identified in previous studies as potential confounders for cognitive impairment were first extracted. Subsequently, all of these variables were included in the logistic regression model. Variables with a p-value greater than 0.2 were then removed through a backward elimination process, allowing only the significant variables to remain in the final model. In the next step, the adjusted logistic regression model was developed based on these key variables.

In the adjusted model, age, BMI, educational level, total cholesterol, physical activity, smoking, diabetes, hypertension, socio-economic index, and dyslipidemia were entered as potential confounde. Multicollinearity was assessed using the Hosmer–Lemeshow test, model goodness-of-fit, and pseudo-R² statistics for the logistic regression models.

### Ethics approval

This study was conducted in accordance with the Declaration of Helsinki. The Research Ethics Committee of Tehran University of Medical Sciences approved the study protocol (Ethical code: IR.TUMS.EMRI.REC.1403.048). All participants provided written informed consent after being fully informed about the procedures involved. For illiterate participants, written informed consent was obtained from their legal guardians.

## Results

### Characteristics of the study population

A total of 2,426 individuals (mean age: 65.34 ± 6.40 years) were included in this analysis, comprising 1,166 males (48.06%) and 1,260 females (51.94%). Age and use of glucose-lowering drugs did not differ between males and females. The mean body mass index (BMI) was 27.52 ± 4.90. Among participants, 20.79% were current smokers, 33.50% were illiterate, and 36.51% had an education at the primary school level. Hypertension and diabetes mellitus were present in 55.92% and 29.40%, respectively. Overall, 1,550 individuals (63.89%) were identified with cognitive impairment, with females significantly higher than in males (P < 0.001) (Table 1).

In males, significant differences between individuals with and without cognitive impairment were observed in age, BMI, socio-economic index, education level, and physical activity. In females, differences were significant between the two groups for age, BMI, smoking status, socio-economic index, education level, and hypertension (Table 1).

Among the biochemical variables, HBA1C and FBS were associated with cognition in males. There was no statistical difference regarding SBP, FBS, and HbA1C results between males and females. None of the assessed novel anthropometric indices showed significant differences between the two groups in either males or females (Table 2). There was a significant association between cognitive impairment and gender in the FAST test (P-value < 0.001), Mini-Cog test (P-value < 0.001), and CFT test (P-value = 0.021).

### The association between anthropometric indices and each cognitive test among males

In the adjusted model, VAI as a continuous variable was significantly associated with cognitive impairment in the FAST test (OR: 1.60, 95% CI: 1.16–2.20; P = 0.004, pseudo-R²: 0.640; Hosmer–Lemeshow test: 0.663). No statistically significant associations were observed between any of the indices and cognitive impairment based on the CFT results (Table 3).

### The association between anthropometric indices and each cognitive test among females

In females, only VAI was significantly associated with cognitive impairment based on the FAST test, with second-quartile participants showing lower odds (OR: 0.58, 95% CI: 0.34–1.00; P = 0.049; pseudo R²: 0.050; Hosmer–Lemeshow test: 0.618) in the adjusted model. TyG and LAP, were not associated with any cognitive test scores in females (Table 4).

### The association between anthropometric indices and cognitive impairment

When all cognitive tests were considered together, none of the anthropometric indices showed a statistically significant association with cognitive impairment in males or females in the adjusted models (Table 5).

## Discussion

In this study, in males, higher VAI was significantly associated with cognitive impairment based on the FAST test. In females, individuals in the second quartile of VAI had significantly lower odds of cognitive impairment in the FAST test. No significant associations were observed between any of the anthropometric indices and cognitive performance in the CFT, and overall, the findings did not reveal a consistent pattern across cognitive domains.

Sex-related variations in education, functional status, and overall health may help explain the pattern of findings in our study. In many aging populations, older females often have lower educational attainment and a higher burden of functional limitations compared with males, both of which are strong determinants of cognitive test performance [37]. These contextual differences may partly underlie the higher prevalence of cognitive impairment observed in females in our sample and may influence how metabolic indices relate to cognitive outcomes. Beyond these social and functional differences, biological mechanisms may also contribute to sex-specific patterns. Estrogen and other sex hormones play a central role in regulating visceral adiposity, adipocyte metabolism, inflammatory responses, and fat-distribution pathways, leading to

Table 1. Demographic characteristics of the study participants.

| Variables | | Total | Males (n = 1166) | | | | Females (n = 1260) | | | | Comparison between male and female |
|---|---|---|---|---|---|---|---|---|---|---|---|
| | | | Impaired cognition (n = 641) | Normal cognition (n = 525) | Total | P-value | Impaired cognition (n = 909) | Normal cognition (n = 351) | Total | P-value | P-value |
| Age, years, (mean± SD) | | 69.34±6.40 | 70.76±7.01 | 68.05±5.32 | 69.54±6.44 | < 0.001 | 69.71±6.60 | 67.74±5.43 | 69.16±6.36 | < 0.001 | 0.146 |
| BMI, kg/m² (mean± SD) | | 27.52±4.90 | 25.96±4.04 | 26.57±3.97 | 26.24±4.02 | 0.01 | 28.47±5.39 | 29.30±5.14 | 28.70±5.34 | 0.014 | < 0.001 |
| Current Smoker | Yes | 504 (20.79) | 159 (24.80) | 113 (21.50) | 272 (23.30) | 0.187 | 181 (19.90) | 51 (14.60) | 232 (18.40) | 0.03 | 0.003 |
| Socio-economic index | Q1 | 71 (2.94) | 9 (1.40) | 5 (1.00) | 14 (1.20) | 0.005 | 46 (5.10) | 11 (3.20) | 57 (4.50) | 0.326 | < 0.001 |
| | Q2 | 115 (4.76) | 27 (4.20) | 7 (1.30) | 34 (2.90) | | 59 (6.50) | 22 (6.30) | 81 (6.50) | | |
| | Q3 | 134 5.54) | 26 (4.10) | 12 (2.30) | 38 (3.30) | | 75 (8.30) | 21 (6.00) | 96 (7.70) | | |
| | Q4 | 216 (8.94) | 34 (5.30) | 42 (8.00) | 76 (6.50) | | 102 (11.30) | 38 (10.90) | 140 (11.20) | | |
| | Q5 | 1881 (77.82) | 544 (85.00) | 457 (87.40) | 1001 (86.10) | | 623 (68.80) | 257 (73.80) | 880 (70.20) | | |
| Education | Illiterate | 800 (33.50) | 166 (25.90) | 47 (9.00) | 213(18.30) | < 0.001 | 477 (52.50) | 110 (31.50) | 587(46.70) | < 0.001 | < 0.001 |
| | Primary school | 885 (36.51) | 266 (41.50) | 144 (27.40) | 410 (35.20) | | 351 (38.60) | 124 (35.50) | 475 (38.70) | | |
| | High school | 218 (8.99) | 66 (10.30) | 78 (14.90) | 144 (12.30) | | 43 (4.70) | 31 (8.90) | 144 (12.30) | | |
| | Diploma | 332 (13.70) | 82 (12.80) | 154 (29.30) | 236 (20.20) | | 29 (3.20) | 67 (19.20) | 236 (20.20) | | |
| | College degree or above | 189 (7.80) | 61 (9.50) | 102 (19.40) | 163 (14.00) | | 9 (1.00) | 17 (4.90) | 163 (14.00) | | |
| Physically active | Yes | 555 (22.90) | 118 (18.40) | 153 (29.10) | 271 (23.20) | < 0.001 | 196 (21.60) | 88 (25.20) | 284 (22.60) | 0.165 | |
| Hypertension | Yes | 1355 (55.92) | 314 (49.00) | 277 (52. 80) | 562 (48.20) | 0.552 | 311 (34.20) | 195 (55.60) | 793 (62.90) | < 0.001 | < 0.001 |
| Type2 Diabetes mellitus | Yes | 714(29.40) | 193 (30.20) | 134 (25. 60) | 327(28.10) | 0.084 | 322 (35.50) | 118 (33.60) | 440(35.00) | 0.521 | < 0.001 |
| Dyslipidemia | Yes | 1773 (73.11) | 413(64.40) | 342 (65. 10) | 755 (64.80) | 0.8 | 738 (71.30) | 280 (79.80) | 1018 (80.90) | 0.543 | < 0.001 |
| Blood pressure lowering agents | Yes | 1282 (95.17) | 287 (92.90) | 233 (94. 00) | 520 (93.40) | 0.614 | 575 (96.50) | 187 (96.40) | 762 (96.50) | 0.956 | |
| Glucose lowering agents | Yes | 663 (92.99) | 162 (92.60) | 112 (94. 10) | 274 (93.20) | 0.605 | 289 (92.00) | 100 (95.20) | 389(92.80) | 0.271 | 0.854 |

Data are presented as mean± standard deviation or number (percent).

P-value: Chi-squared test, Independent Samples T-Test.

BMI: body mass index;

**Table 2. Biochemical characteristics of the study participants.**

| Variables | | Total | Males (n=1166) | | | | Females (n=1260) | | | | Comparison between male and female P-value |
|---|---|---|---|---|---|---|---|---|---|---|---|
| | | | Impaired cognition (n=641) | Normal cognition (n=525) | Total | P-value | Impaired cognition (n=909) | Normal cognition (n=351) | Total | P-value | |
| WC,cm, (mean± SD) | | 98.72±12.02 | 96.68±11.57 | 97.58±10.78 | 97.08±11.22 | 0.172 | 100.14±12.84 | 100.47±11.67 | 100.23±12.52 | 0.673 | <0.001 |
| SBP,mmHg, (mean± SD) | | 140.59±20.43 | 141.18±21.77 | 140.04±18.92 | 140.67±20.53 | 0.346 | 140.48±20.57 | 140.62±19.76 | 140.52±20.34 | 0.915 | 0.328 |
| DBP,mmHg, (mean± SD) | | 81.55±9.51 | 81.98±9.58 | 82.64±9.75 | 82.28±9.58 | 0.242 | 80.77±9.43 | 81.13±9.24 | 80.87±9.40 | 0.549 | <0.001 |
| FBS,mg/dl, (mean± SD) | | 106.18±42.63 | 107.04±44.25 | 101.57±34.70 | 104.58±40.31 | 0.021 | 108.28±45.88 | 106.10±41.27 | 107.67±44.94 | 0.438 | 0.074 |
| HbA1C, %, (mean± SD) | | 5.67±1.56 | 5.76±1.66 | 5.55±1.41 | 5.66±1.55 | 0.022 | 5.69±1.59 | 5.68±1.52 | 5.68±1.57 | 0.93 | 0.720 |
| Total cholesterol, mg/dl,(mean± SD) | | 182.11±44.29 | 172.79±40.93 | 173.94±40.75 | 173.31±40.84 | 0.633 | 189.82±45.78 | 191.36±45.91 | 190.25±45.80 | 0.593 | <0.001 |
| LDL-C, mg/dl, (mean± SD) | | 109.33±37.83 | 104.66±34.80 | 104.46±34.87 | 104.57±34.82 | 0.924 | 113.83±40.03 | 113.48±39.72 | 113.73±39.93 | 0.89 | <0.001 |
| HDL-C, mg/dl | | 45.92±11.25 | 43.13±10.24 | 43.04±9.10 | 43.08±10.12 | 0.875 | 48.19±11.31 | 49.46±12.31 | 48.55±11.60 | 0.082 | <0.001 |
| Triglycerides, mg/dl | | 135.87±70.48 | 127.00±65.60 | 134.25±71.27 | 130.27±68.28 | 0.071 | 139.91±69.54 | 144.02±78.35 | 141.06±72.09 | 0.365 | <0.001 |
| TyG index | | 8.72±0.61 | 8.72±0.61 | 8.67±0.57 | 8.66±0.61 | 0.535 | 8.78±0.60 | 8.77±0.61 | 8.78±0.61 | 0.867 | <0.001 |
| TyG index (quartiles) | Q1 | 466 (19.20) | 169 (25.60) | 115 (21.90) | 284 (24.40) | 0.198 | 224 (24.70) | 88 (25.10) | 312 (24.80) | 0.891 | |
| | Q2 | 737(30.40) | 164(25.60) | 143(27.20) | 296(25.40) | | 231(25.40) | 82(23.40) | 313(24.90) | | |
| | Q3 | 613 (25.30) | 156 (24.30) | 138 (26.30) | 290 (24.90) | | 232 (25.60) | 94 (26.80) | 326 (25.90) | | |
| | Q4 | 609 (25.10) | 157 (24.50) | 129 (24.60) | 296 (25.40) | | 221 (24.30) | 87 (24.80) | 308(24.50) | | |
| LAP | – | | 47.34±33.04 | 50.76±33.64 | 48.88±33.34 | 0.082 | 68.17±42.74 | 68.17±42.75 | 68.61±42.74 | 0.559 | <0.001 |
| LAP (quartiles) | Q1 | – | 174 (27.10) | 116 (22.10) | 290 (24.90) | 0.841 | 235(25.90) | 79(22.60) | 314 (25.00) | 0.579 | |
| | Q2 | – | 162(25.30) | 131(25.00) | 293(25.10) | | 220(24.20) | 95(27.20) | 315(25.00) | | |
| | Q3 | – | 152 (23.70) | 139 (26.50) | 291 (25.00) | | 227 (25.00) | 88 (25.20) | 314 (25.00) | | |
| | Q4 | – | 153 (23.90) | 139 (26.50) | 292 (25.00) | | 227 (25.00) | 87 (24.90) | 315(25.00) | | |
| VAI | – | | 2.07±1.48 | 2.18±1.55 | 2.12±1.51 | 0.229 | 2.95±2.21 | 2.95±2.21 | 2.95±2.25 | 0.969 | <0.001 |
| VAI (quartiles) | Q1 | – | 164 (25.60) | 125 (23.80) | 289 (24.80) | 0.2 | 223(24.50) | 90(25.80) | 313(24.90) | 0.946 | |
| | Q2 | – | 164(25.60) | 130(24.80) | 294(25.20) | | 227(25.00) | 89(25.50) | 316(25.10) | | |
| | Q3 | – | 156 (24.30) | 134 (25.50) | 290 (24.90) | | 229 (25.20) | 86 (24.60) | 315(25.00) | | |
| | Q4 | – | 157 (24.50) | 136 (25.90) | 293 (25.10) | | 230 (25.30) | 84 (24.10) | 314(25.00) | | |

Data are presented as mean± standard deviation or number (percent).

P-value: Chi-squared test, Independent Samples T-Test.

WC: waist circumference; SBP: systolic blood pressure; DBP: diastolic blood pressure; FBS: fasting blood sugar; LDL-C: low-density lipoprotein cholesterol; HDL-C: high-density lipoprotein cholesterol; TyG: triglyceride glucose; LAP: lipid accumulation products; VAI: visceral adiposity index.

**Table 3. The association between anthropometric indices and each cognitive test among males.**

| | Mini-cog: Crude OR (95% CI) | Mini-cog: Crude p-value | Mini-cog: Adjusted* OR (95% CI) | Mini-cog: Adjusted* p-value | Mini-cog: Pseudo | Mini-cog: *HL | FAST: Crude OR (95% CI) | FAST: Crude p-value | FAST: Adjusted* OR (95% CI) | FAST: Adjusted* p-value | FAST: Pseudo | FAST: *HL | CFT: Crude OR (95% CI) | CFT: Crude p-value | CFT: Adjusted* OR (95% CI) | CFT: Adjusted* p-value | CFT: Pseudo | CFT: *HL |
|---|---|---|---|---|---|---|---|---|---|---|---|---|---|---|---|---|---|---|
| TyG index (continuous) | 1.459 (0.809–2.631) | 0.209 | 1.475 (0.706–3.080) | 0.301 | 0.159 | 0.327 | 0.653 (0.253–1.682) | 0.377 | 0.629 (0.222–1.785) | 0.383 | 0.065 | 0.158 | 0.897 (0.426–1.887) | 0.755 | 0.810 (0.355–1.845) | 0.615 | 0.031 | 0.359 |
| TyG index (quartiles) | | | | | | | | | | | | | | | | | | |
| Q1 | Reference | Reference | Reference | Reference | Reference | Reference | Reference | Reference | Reference | Reference | Reference | Reference | Reference | Reference | Reference | Reference | Reference | Reference |
| Q2 | 0.661 (0.416–1.050) | 0.08 | 0.621 (0.367–1.053) | 0.077 | 0.109 | 0.146 | 0.829 (0.417–1.646) | 0.591 | 0.790 (0.389–1.607) | 0.516 | 0.051 | 0.171 | 1.155 (0.650–2.051) | 0.624 | 1.126 (0.623–2.035) | 0.694 | 0.029 | 0.361 |
| Q3 | 0.600 (0.315–1.145) | 0.121 | 0.556 (0.267–1.159) | 0.117 | 0.163 | 0.301 | 0.719 (0.270–1.916) | 0.509 | 0.640 (0.234–1.749) | 0.384 | 0.071 | 0.164 | 1.329 (0.600–2.947) | 0.483 | 1.307 (0.580–2.945) | 0.518 | 0.042 | 0.341 |
| Q4 | 0.668 (0.273–1.730) | 0.426 | 0.493 (0.173–1.400) | 0.184 | 0.181 | 0.208 | 0.731 (0.175–3.048) | 0.667 | 0.581 (0.135–2.503) | 0.466 | 0.89 | 0.201 | 1.932 (0.610–6.123) | 0.263 | 1.837 (0.569–5.930) | 0.309 | 0.057 | 0.290 |
| LAP (continuous) | 0.991 (0.982–1.001) | 0.083 | 0.988 (0.976–1.002) | 0.083 | 0.160 | 0.322 | 0.990 (0.975–1.005) | 0.2 | 0.984 (0.965–1.002) | 0.089 | 0.064 | 0.657 | 1.000 (0.988–1.013) | 0.997 | 1.005 (0.990–1.021) | 0.497 | 0.030 | 0.354 |
| LAP (quartiles) | | | | | | | | | | | | | | | | | | |
| Q1 | Reference | Reference | Reference | Reference | Reference | Reference | Reference | Reference | Reference | Reference | Reference | Reference | Reference | Reference | Reference | Reference | Reference | Reference |
| Q2 | 0.743 (0.501–1.103) | 0.141 | 0.644 (0.397–1.044) | 0.074 | 0.103 | 0.372 | 1.718 (0.949–3.108) | 0.074 | 1.383 (0.714–2.677) | 0.336 | 0.264 | 0.357 | 0.847 (0.522–1.376) | 0.503 | 0.989 (0.579–1.689) | 0.966 | 0.030 | 0.354 |
| Q3 | 0.816 (0.481–1.385) | 0.452 | 0.910 (0.473–1.753) | 0.778 | 0.212 | 0.222 | 1.133 (0.493–2.603) | 0.769 | 0.892 (0.351–2.269) | 0.811 | 0.174 | 0.387 | 0.613 (0.319–1.176) | 0.141 | 0.838 (0.405–1.734) | 0.634 | 0.030 | 0.354 |
| Q4 | 1.054 (0.481–2.312) | 0.895 | 1.083 (0.424–2.767) | 0.868 | 0.290 | 0.201 | 2.127 (0.646–7.003) | 0.215 | 1.665 (0.450–6.157) | 0.445 | 0.214 | 0.357 | 0.434 (0.160–1.180) | 0.102 | 0.616 (0.212–1.788) | 0.373 | 0.030 | 0.354 |
| VAI (continuous) | 0.936 (0.771–1.136) | 0.504 | 1.047 (0.822–1.334) | 0.710 | 0.159 | 0.326 | *1.343 (1.013–1.780) | 0.04 | *1.601 (1.164–2.203) | 0.004 | 0.640 | 0.663 | 0.982 (0.758–1.272) | 0.890 | 0.956 (0.706–1.293) | 0.770 | 0.031 | 0.350 |
| VAI (quartiles) | | | | | | | | | | | | | | | | | | |
| Q1 | Reference | Reference | Reference | Reference | Reference | Reference | Reference | Reference | Reference | Reference | Reference | Reference | Reference | Reference | Reference | Reference | Reference | Reference |
| Q2 | 1.083 (0.711–1.643) | 0.709 | 0.997 (0.606–1.640) | 0.991 | 0.201 | 0.286 | 1.030 (0.548–1.938) | 0.927 | 1.174 (0.558–2.343) | 0.65 | 0.314 | 0.383 | 1.491 (0.889–2.500) | 0.130 | 1.486 (0.851–2.594) | 0.161 | 0.029 | 0.184 |

*(Continued)*

**Table 3.** (Continued)

| | Mini-cog test score | | | | | | FAST test score | | | | | | CFT test score | | | | | |
| | Crude | | Adjusted* | | | | Crude | | Adjusted* | | | | Crude | | Adjusted* | | | |
| | OR (95% confidence interval) | p-value | OR (95% confidence interval | p-value | Pseudo | *HL | OR (95% confidence interval | p-value | OR (95% confidence interval) | p-value | Pseudo | *HL | OR (95% confidence interval | p-value | OR (95% confidence interval | p-value | Pseudo | *HL |
|---|---|---|---|---|---|---|---|---|---|---|---|---|---|---|---|---|---|---|
| Q3 | 1.326 (0.784–2.244) | 0.293 | 1.084 (0.558–2.107) | 0.811 | 0.195 | 0.346 | 1.714 (0.801–3.667) | 0.165 | 1.987 (0.813–4.856) | 0.132 | 0.164 | 0.363 | 1.881(0.985–3.592) | 0.056 | 1.799 (0.862–3.754) | 0.118 | 0.030 | 0.172 |
| Q4 | 1.490 (0.721–3.081) | 0.282 | 1.218 (0.492–1.013) | 0.670 | 0.147 | 0.304 | 0.837 (0.275–2.549) | 0.754 | 0.942 (0.264–3.361) | 0.926 | 0.314 | 0.263 | 1.306(0.515–3.313) | 0.574 | 1.125 (0.396–3.196) | 0.824 | 0.190 | 0.350 |

Adjusted for age, BMI, educational level, total cholesterol, physical activity, smoking, diabetes, hypertension, socio-economic index, dyslipidemia

Data are presented as OR (95% confidence interval). p<0.05.

TyG: triglyceride glucose; VAI: visceral adiposity index; LAP: lipid accumulation product; FAST: functional assessment staging test; CFT: Category fluency test.

*HL.: Hosmer–Lemeshow Test

**Table 4. The association between anthropometric indices and each cognitive test among females.**

| | Mini-cog test score | | | | | | FAST test score | | | | | | CFT score | | | | | |
|---|---|---|---|---|---|---|---|---|---|---|---|---|---|---|---|---|---|---|
| | Crude | | Adjusted* | | Pseudo | *HL | Crude | | Adjusted* | | Pseudo | *HL | Crude | | Adjusted* | | Pseudo | *HL |
| | OR (95% confidence interval) | p-value | OR (95% confidence interval) | p-value | | | OR (95% confidence interval) | p-value | OR (95% confidence interval) | p-value | | | OR (95% confidence interval) | p-value | OR (95% confidence interval) | p-value | | |
| TyG index (continuous) | 1.304 (0.756–2.247) | 0.34 | 1.217 (0.677–2.186) | 0.512 | 0.145 | 0.392 | 1.570 (0.823–2.995) | 0.171 | 1.601 (0.824–3.110) | 0.165 | 0.072 | 0.745 | 1.226 (0.674–2.229) | 0.505 | 1.153 (0.620–2.144) | 0.653 | 0.028 | 0.376 |
| TyG index (quartiles) | | | | | | | | | | | | | | | | | | |
| Q1 | Reference | Reference | Reference | Reference | Reference | Reference | Reference | Reference | Reference | Reference | Reference | Reference | Reference | Reference | Reference | Reference | Reference | Reference |
| Q2 | 0.984 (0.634–1.527) | 0.94 | 1.027 (0.635–1.661) | 0.913 | 0.235 | 0.292 | 1.033 (0.602–1.772) | 0.907 | 1.055 (0.604–1.840) | 0.851 | 0.262 | 0.445 | 0.794 (0.478–1.318) | 0.372 | 0.738 (0.438–1.244) | 0.255 | 0.126 | 0.276 |
| Q3 | 0.683 (0.376–1.240) | 0.21 | 0.744 (0.391–1.415) | 0.367 | 0.105 | 0.408 | 0.756 (0.362–1.578) | 0.457 | 0.759 (0.355–1.623) | 0.477 | 0.172 | 0.365 | 1.048 (0.534–2.055) | 892 | 1.093 (0.546–2.188) | 0.801 | 0.228 | 0.176 |
| Q4 | 0.688 (0.290–1.628) | 0.394 | 0.704 (0.279–1.781) | 0.459 | 0.125 | 0.298 | 0.745 (0.261–2.126) | 0.582 | 0.719 (0.244–2.117) | 0.55 | 0.072 | 0.445 | 0.739 (0.279–1.957) | 0.543 | 0.773 (0.284–2.203) | 0.613 | 0.348 | 0.171 |
| LAP (continuous) | 0.994 (0.987–1.001) | 0.071 | 0.995 (0.986–1.003) | 0.208 | 0.104 | 0.414 | 0.994 (0.985–1.003) | 0.188 | 0.998 (0.987–1.008) | 0.655 | 0.073 | 0.627 | 1.003 (0.995–1.010) | 0.486 | 1.007 (0.998–1.016) | 0.144 | 0.031 | 0.351 |
| LAP (quartiles) | | | | | | | | | | | | | | | | | | |
| Q1 | Reference | Reference | Reference | Reference | Reference | Reference | Reference | Reference | Reference | Reference | Reference | Reference | Reference | Reference | Reference | Reference | Reference | Reference |
| Q2 | 0.824 (0.554–1.226) | 0.301 | 0.960 (0.612–1.506) | 0.895 | 0.051 | 0.114 | 0.879 (0.537–1.438) | 0.606 | 0.990 (0.583–1.680) | 0.97 | 0.173 | 0.327 | 0.733 (0.468–1.146) | 0.173 | 0.842 (0.519–1.368) | 0.488 | 0.038 | 0.301 |
| Q3 | 0.911 (0.547–1.517) | 0.721 | 1.334 (0.739–2.409) | 0.339 | 0.121 | 0.211 | 0.968 | 0.92 | 1.272 (0.637–2.538) | 0.495 | 0.103 | 0.348 | 0.558 (0.314–0.991) | 0.046 | 0.715 (0.379–1.350) | 0.301 | 0.051 | 0.371 |
| Q4 | 1.112 (0.531–2.331) | 0.778 | 1.821 (0.794–4.177) | 0.157 | 0.062 | 0.301 | 1.036 | 0.94 | 1.439 (0.541–3.380) | 0.466 | 0.111 | 0.318 | 0.414 (0.181–0.947) | 0.037 | 0.517 (0.212–1.260) | 0.147 | 0.048 | 0.401 |
| VAI (continuous) | 0.945 (0.849–1.052) | 0.304 | 0.921 (0.814–1.041) | 0.188 | 0.104 | 0.412 | 0.962 (0.829–1.117) | 0.612 | 0.911 (0.768–1.080) | 0.281 | 0.073 | 0.635 | 1.052 (0.942–1.175) | 0.368 | 1.022 (0.902–1.157) | 0.736 | 0.031 | 0.360 |
| VAI (quartiles) | | | | | | | | | | | | | | | | | | |
| Q1 | Reference | Reference | Reference | Reference | Reference | Reference | Reference | Reference | Reference | Reference | Reference | Reference | Reference | Reference | Reference | Reference | Reference | Reference |
| Q2 | 1.465 (0.984–2.182) | 0.06 | 1.089 (0.692–1.713) | 0.714 | 0.187 | 0.112 | 0.689 (0.417–1.136) | 0.144 | *0.583 (0.341–0.997) | 0.049 | 0.050 | 0.618 | 0.855 (0.540–1.355) | 0.505 | 0.731 (0.446–1.196) | 0.212 | 0.061 | 0.160 |

*(Continued)*

**Table 4.** (Continued)

| | Mini-cog test score | | | | | | FAST test score | | | | | | CFT score | | | | | |
| | Crude | | Adjusted* | | | | Crude | | Adjusted* | | | | Crude | | Adjusted* | | | |
| | OR (95% confidence interval) | p-value | OR (95% confidence interval) | p-value | Pseudo | *HL | OR (95% confidence interval) | p-value | OR (95% confidence interval) | p-value | Pseudo | *HL | OR (95% confidence interval) | p-value | OR (95% confidence interval) | p-value | Pseudo | *HL |
|---|---|---|---|---|---|---|---|---|---|---|---|---|---|---|---|---|---|---|
| Q3 | *1.738 (1.065–2.837) | 0.027 | 1.072 (0.596–1.927) | 0.817 | 0.104 | 0.202 | 1.122 (0.613–2.056) | 0.708 | 0.799 (0.403–1.583) | 0.52 | 0.172 | 0.335 | 1.077 (0.615–1.884) | 0.796 | 0.735 (0.391–1.382) | 0.339 | 0.116 | 0.260 |
| Q4 | *2.402 (1.267–4.554) | 0.007 | 1.370 (0.646–2.904) | 0.411 | 0.131 | 0.302 | 1.385 (0.619–3.098) | 0.428 | 0.906 (0.369–2.223) | 0.829 | 0.278 | 0.135 | 1.275 (0.627–2.596) | 502 | 0.801 (0.361–1.755) | 0.585 | 0.178 | 0.118 |

*Adjusted for age, BMI, educational level, total cholesterol, physical activity, smoking, diabetes, hypertension, socio-economic index, dyslipidemia

Data are presented as OR (95% confidence interval). $p<0.05$.

TyG: triglyceride glucose; VAI: visceral adiposity index; LAP: lipid accumulation product; FAST: functional assessment staging test; AFT: category fluency test.

*HL: Hosmer–Lemeshow Test

**Table 5. The association between anthropometric indices and cognitive impairment.**

| | Males | | | | | | Females | | | | | |
| --- | --- | --- | --- | --- | --- | --- | --- | --- | --- | --- | --- | --- |
| | Crude | | Adjusted* | | | | Crude | | Adjusted* | | | |
| | OR (95% confidence interval | p-value | OR (95% confidence interval | p-value | Pseudo | *HL | OR (95% confidence interval | p-value | OR (95% confidence interval | p-value | Pseudo | *HL |
| TyG index (continuous) | 1.138 (0.630–2.057) | 0.668 | 1.055 (0.518–2.149) | 0.882 | 0.114 | 0.288 | 1.177 (0.653–2.121) | 0.587 | 0.980 (0.505–1.903) | 0.952 | 0.303 | 0.116 |
| TyG index (quartiles) | | | | | | | | | | | | |
| Q1 | Reference | Reference | Reference | Reference | Reference | Reference | Reference | Reference | Reference | Reference | Reference | Reference |
| Q2 | 0.720 (0.453–1.144) | 0.164 | 0.672 (0.405–1.114) | 0.123 | 0.224 | 0.380 | 1.062 (0.659–1.713) | 0.804 | 1.107 (0.660–1.857) | 0.7 | 0.289 | 0.220 |
| Q3 | 0.789 (0.414–1.504) | 0.471 | 0.742 (0.367–1.501) | 0.406 | 0.311 | 0.181 | 0.781 (0.410–1.488) | 0.453 | 0.821 (0.413–1.631) | 0.573 | 0.151 | 0.202 |
| Q4 | 1.044 (0.414–2.635) | 0.927 | 0.814 (0.279–1.228) | 0.689 | 0.311 | 0.112 | 0.732 (0.289–1.857) | 0.511 | 0.689 (0.255–1.860) | 0.463 | 0.108 | 0.0288 |
| LAP (continuous) | 0.997 (0.988–1.006) | 0.546 | 1.000 (0.987–1.012) | 0.936 | 0.247 | 0.192 | 0.997 (0.990–1.005) | 0.501 | 1.000 (0.926–1.009) | 0.926 | 0.290 | 0.148 |
| LAP (quartiles) | | | | | | | | | | | | |
| Q1 | Reference | Reference | Reference | Reference | Reference | Reference | Reference | Reference | Reference | Reference | Reference | Reference |
| Q2 | 0.825 (0.556–1.224) | 0.339 | 0.894 (0.566–1.412) | 0.630 | 0.378 | 0.119 | 0.704 (0.458–1.082) | 0.109 | 0.804 (0.494–1.310) | 0.382 | 0.105 | 0.1271 |
| Q3 | 0.699 (0.414–1.178) | 0.179 | 0.918 (0.494–1.704) | 0.786 | 0.288 | 0.157 | 0.773 (0.445–1.344) | 0.362 | 1.148 (0.604–2.184) | 0.674 | 0.219 | 0.105 |
| Q4 | 0.675 (0.313–1.459) | 0.318 | 0.852 (0.352–2.063) | 0.723 | 0.289 | 0.123 | 0.850 (0.383–1.886) | 0.69 | 1.413 (0.574–3.477) | 0.452 | 0.308 | 0.109 |
| VAI (continuous) | 0.938 (0.776–1.135) | 0.512 | 0.982 (0.782–1.233) | 0.873 | 0.354 | 0.138 | 0.984 (0.879–1.101) | 0.775 | 0.970 (0.852–1.104) | 0.643 | 0.203 | 0.119 |
| VAI (quartiles) | | | | | | | | | | | | |
| Q1 | Reference | Reference | Reference | Reference | Reference | Reference | Reference | Reference | Reference | Reference | Reference | Reference |
| Q2 | 1.282 (0.844–1.945) | 0.244 | 1.211 (0.754–1.944) | 0.429 | 0.48 | 0.318 | 1.232 (0.801–1.897) | 0.342 | 0.926 (0.562–1.524) | 0.761 | 0.175 | 0.148 |
| Q3 | *1.368 (0.812–2.305) | 0.239 | 1.164 (0.618–2.191) | 0.638 | 0.288 | 0.203 | 1.476 (0.869–2.507) | 0.15 | 0.924 (0.481–1.772) | 0.811 | 0.176 | 0.203 |
| Q4 | *1.454 (0.709–2.983) | 0.307 | 1.154 (0.488–2.728) | 0.745 | 0.298 | 0.189 | 1.735 (0.773–3.448) | 0.116 | 0.974 (0.426–2.229) | 0.95 | 0.180 | 0.119 |

*Adjusted for age, BMI, educational level, total cholesterol, physical activity, smoking, diabetes, hypertension, socio-economic index, dyslipidemia

Data are presented as OR (95% confidence interval). p<0.05.

TyG: triglyceride glucose; VAI: visceral adiposity index; LAP: lipid accumulation product.

*HL: Hosmer–Lemeshow Test.

distinct metabolic profiles in males and females. These pathways are themselves implicated in cognitive aging and may modulate susceptibility to metabolic risk factors [38].

There has been ongoing interest in identifying both the underlying mechanisms of cognitive impairment and reliable measures to predict its onset. Although some studies did not find significant differences in IR between individuals with

and without cognitive impairment [39], there is growing support for its role in contributing to cognitive decline [40]. Proposed pathophysiological mechanisms include enlarged perivascular spaces, altered cerebral glucose metabolism, and increased regional deficits in synaptic connectivity [41]. Most studies assess insulin resistance using the HOMA-IR, which remains a widely accepted standard [41]. However, recent research has introduced novel measures that may offer simpler or more practical alternatives for evaluating the association between insulin resistance and cognitive function [42].

The TyG index has emerged as a novel and simple surrogate marker for IR, demonstrating high sensitivity and specificity [43]. It has been proposed as a practical indicator for various conditions, including microvascular and macrovascular complications of diabetes and cardiovascular diseases [43]. More recently, TyG has been studied in relation to cognitive impairment and dementia, including Alzheimer's and vascular dementia [44]. Although several studies support this association [8], others have found no significant or only borderline associations [45,46]. In the present study, no association was observed. Several factors may explain this inconsistency: TyG reflects peripheral rather than central IR [47]; unlike HOMA-IR, which is insulin-based and a closer proxy of systemic insulin signaling that may better associate to brain insulin pathways [48]; it may be influenced by diet and short-term metabolic changes [49]; and variations in cognitive assessment methods and study designs may contribute to inconsistent findings across the literature [8]. Therefore, although TyG is convenient and promising, it may not yet be a reliable predictor of cognitive impairment compared to HOMA-IR. Also, we did not find sex-specific association in our TyG results which may indicate that this index may not reflect the metabolic processes that diverge between males and females as they age.

There are mixed findings regarding the association between the VAI and cognitive impairment across the literature. Several studies report that higher visceral fat is associated with poorer cognitive performance. For instance, one study suggested that early detection and management of visceral obesity may help prevent Alzheimer's disease [50]. Another study indicated that increased BMI and visceral fat were likely contributing factors to cognitive decline [51]. Similarly, a study in older adults with type 2 diabetes found that higher VAI was significantly associated with a greater risk of mild cognitive impairment [52]. In contrast, one study reported that although VAI was inversely associated with working memory (as measured by digit span), it showed no significant association with cognitive tests like MMSE or MoCA [53]. Interestingly, a longitudinal cohort study reported that individuals with higher VAI experienced slower cognitive decline and performed better in episodic memory [22]. In our study, VAI showed a statistically significant association with cognitive impairment only in the FAST test and primarily in males; however, the magnitude of this association was modest, suggesting limited practical importance in clinical contexts without additional risk factors. In contrast, among females, VAI exhibited a protective effect in the second quartile that was also modest in scale and may not translate to substantial real-world impact on cognitive outcomes. These sex-specific patterns, risk-increasing in males but protective in females, may reflect a combination of metabolic differences and variations in cognitive and functional status between sexes, though the modest effect sizes underscore the need for further research to assess their broader applicability.

LAP has also shown inconsistent associations with cognitive impairment across studies. A cross-sectional study of 5,542 normal-weight, hypertensive Chinese adults reported that higher LAP was independently associated with better cognitive performance, as measured by MMSE scores [54]. In contrast, a study of adults with type 2 diabetes found significantly higher LAP levels among those with mild cognitive impairment, suggesting a detrimental association [55]. Furthermore, another study reported that higher LAP predicted cognitive impairment specifically in normotensive women, underscoring that the association between LAP and cognition may be influenced by sex and blood pressure status [56]. It is worth noting that VAI and LAP share common components, VAI is calculated using BMI and triglycerides, both of which also influence LAP. This overlap may introduce multicollinearity and complicate the interpretation of their independent associations with cognition, so future studies should test for collinearity or use alternative strategies

We believe that the association between cognitive function and novel anthropometric indices such as LAP, TyG, or VAI is not purely linear or unidirectional but rather part of a complex, multidimensional network of interactions. Insulin resistance may play a role, but other metabolic, vascular, inflammatory, oxidative stress, and adipokine-related pathways are

also likely involved. Future studies should investigate these associations to better understand underlying mechanisms and identify opportunities for early intervention.

This study has several limitations that should be acknowledged. First, its cross-sectional design limits the ability to draw causal inferences between the indices and mild cognitive impairment, as only associations can be identified. Second, although we employed multiple tests to evaluate cognitive impairment in older adults, these assessments may not fully capture the complexity of cognitive function. The FAST is a staging instrument that emphasizes functional decline across dementia stages rather than direct cognitive testing and focuses on functional abilities rather than cognitive testing; it is intended to map functional change over the course of dementia [31]. The Mini-Cog is a practical, brief screening tool with good case-finding utility for dementia, but its clock-drawing and recall components are susceptible to cultural and educational influences and may under detect early impairments in some populations [33]. The CFT probes semantic verbal fluency and correlates with broader neuropsychological batteries, yet performance is strongly affected by education, language proficiency, and cultural background and can vary with acute factors such as fatigue or anxiety; normative and validation studies therefore recommend local norms or adjustment for educational attainment [30]. Third, the use of cognitive tests other than widely established tools like the MMSE may hinder comparability with other studies. Fourth, we did not include reference measures such as HOMA-IR to validate the indices against established standards for assessing insulin resistance. Fifth, potential confounding factors such as medication use were not accounted for, which may have influenced the observed associations. Sixth, although some statistically significant associations were observed, the odds ratios were close to 1.0, suggesting limited biological and clinical relevance and warranting cautious interpretation. Finally, residual confounding from unmeasured variables, including diet quality, APOE genotype, and overlapping components between VAI and LAP (e.g., BMI and triglycerides), cannot be excluded.

## Conclusion

In this large cross-sectional study of Iranian older adults, novel anthropometric indices (TyG, LAP, VAI) were not consistently associated with cognitive impairment; however, some sex-specific associations were observed for VAI. These sex- and test-specific differences suggest the need for longitudinal studies to clarify causal pathways.

### Author contributions

**Conceptualization:** Mahnaz Pejman Sani.

**Formal analysis:** Negar Asaad Sajadi.

**Methodology:** Farshad Sharifi, Kazem Khalagi.

**Project administration:** Bagher Larijani, Iraj Nabipour.

**Supervision:** Noushin Fahimfar, Mahnaz Pejman Sani.

**Validation:** Kazem Khalagi.

**Writing – original draft:** Shervin Mossavarali.

**Writing – review & editing:** Shahrzad Mohseni, Mohammadreza Mohajeri-Tehrani.

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
