## [Decision Letter · Decision Letter 0]

17 Nov 2025

PONE-D-25-56488The Association between Novel Anthropometric Indices and Cognitive Function in Iranian Older Adults: Bushehr Elderly Health (BEH) ProgramPLOS ONE

Dear Dr. Pejman Sani,

Thank you for submitting your manuscript to PLOS ONE. After careful consideration, we feel that it has merit but does not fully meet PLOS ONE’s publication criteria as it currently stands. Therefore, we invite you to submit a revised version of the manuscript that addresses the points raised during the review process.

If applicable, we recommend that you deposit your laboratory protocols in protocols.io to enhance the reproducibility of your results. Protocols.io assigns your protocol its own identifier (DOI) so that it can be cited independently in the future. For instructions see: https://journals.plos.org/plosone/s/submission-guidelines#loc-laboratory-protocols. Additionally, PLOS ONE offers an option for publishing peer-reviewed Lab Protocol articles, which describe protocols hosted on protocols.io. Read more information on sharing protocols at . Additionally, PLOS ONE offers an option for publishing peer-reviewed Lab Protocol articles, which describe protocols hosted on protocols.io. Read more information on sharing protocols at https://plos.org/protocols?utm_medium=editorial-email&utm_source=authorletters&utm_campaign=protocols..

We look forward to receiving your revised manuscript.

Kind regards,

Amin Mansoori

Academic Editor

PLOS ONE

Journal Requirements:

3. In this instance it seems there may be acceptable restrictions in place that prevent the public sharing of your minimal data. However, in line with our goal of ensuring long-term data availability to all interested researchers, PLOS’ Data Policy states that authors cannot be the sole named individuals responsible for ensuring data access (http://journals.plos.org/plosone/s/data-availability#loc-acceptable-data-sharing-methods).

**Additional Editor Comments:**

Dear Authors

This manuscript has several strengths, most notably its large sample size and standardized measurement approach. However, it requires significant revision to meet the journal's publication standards. Major concerns center on statistical methods—specifically the use of stepwise regression and unclear model diagnostics—as well as the justification for the cognitive impairment definition and adherence to the mandatory open-data policy. We encourage the authors to carefully revise the manuscript to fully address the reviewers' detailed comments.

Best Regards

Reviewers' comments:

Reviewer's Responses to Questions

**Comments to the Author**

1. Is the manuscript technically sound, and do the data support the conclusions?

Reviewer #1: Yes

Reviewer #2: Partly

Reviewer #3: Yes

2. Has the statistical analysis been performed appropriately and rigorously? 

Reviewer #1: Yes

Reviewer #2: No

Reviewer #3: Yes

3. Have the authors made all data underlying the findings in their manuscript fully available?

Reviewer #1: Yes

Reviewer #2: No

Reviewer #3: No

4. Is the manuscript presented in an intelligible fashion and written in standard English?

Reviewer #1: No

Reviewer #2: Yes

Reviewer #3: Yes

5. Review Comments to the Author

Reviewer #1: 1. Some sentences are long and could be broken into shorter sentences for readability.

2. References to prevalence in Iran could include more recent statistics if available.

3. Consider emphasizing why these indices are practical alternatives to HOMA-IR (cost and accessibility) a bit earlier to strengthen the rationale.

4. Consider summarizing borderline associations separately from statistically significant results to improve clarity.

5. Some sentences are repetitive (e.g., noting no association for TyG and LAP in multiple places). Consolidating these points could improve readability.

6. Clarify whether quartile-based analyses were adjusted for multiple comparisons.

Reviewer #2: •The study design is cross-sectional, yet several statements imply causality (e.g., “interplay between metabolic indices and cognitive function”). These should be rephrased to reflect association, not cause–effect.

•Some methodological details are insufficient: the phrase “phase 2 of baseline measurements” is ambiguous; it should clearly indicate whether this was a second wave of data collection or a subsample of a longitudinal cohort.

•Definitions of cognitive impairment (based on impairment in ≥1 of 3 tests) may overestimate prevalence; justification or sensitivity analysis using stricter criteria is needed.

•Despite reporting significant associations, the biological and clinical relevance of observed odds ratios (close to 1.0) is minimal and should be interpreted cautiously.

•The manuscript states that covariates were selected using backward stepwise regression, which can lead to unstable models and overfitting. Covariate inclusion should be theory-driven or justified by prior evidence.

•The paper does not describe how missing data were handled. State whether listwise deletion, imputation, or other methods were used.

•No mention of multicollinearity testing, goodness-of-fit (e.g., Hosmer–Lemeshow test), or sensitivity analyses is provided.

•Effect sizes and confidence intervals are reported but should be discussed in terms of practical importance.

•The decision to stratify by sex is reasonable, but the rationale for doing so should be stated explicitly and, ideally, supported by a statistical test for interaction.

•The manuscript is understandable, but several sentences are long, redundant, or grammatically inconsistent.

•Minor typographical errors occur (“demantia,” “measuremnts”).

•Paragraphs in the Discussion are overly descriptive; the narrative should be condensed and more analytical.

•Ensure consistent formatting of tables, units (mmol/L vs mg/dL), and reference style.

A professional English-language edit is recommended before resubmission.

1.Clarify the study design and population. Clearly describe what “phase 2 of baseline” refers to and specify inclusion/exclusion criteria.

2.Refine the definition of cognitive impairment. Provide justification for your threshold (impairment in ≥1 test) and test an alternative stricter definition in sensitivity analysis.

3.Re-examine model specification. Replace stepwise regression with theory-driven models or demonstrate that results are robust to different covariate sets.

4.Address data handling. Explain how missing values were treated and whether multicollinearity was assessed.

5.Add regression diagnostics. Include measures such as pseudo-R² and goodness-of-fit statistics.

6.Report and interpret effect sizes more cautiously. ORs near 1.0 may be statistically significant but clinically negligible.

7.Revise discussion and conclusion. Avoid causal language; discuss mechanisms and confounders (sex hormones, fat distribution, inflammation).

8.Ensure open data compliance. Deposit anonymized dataset in a public repository and update the Data Availability Statement.

9.Language and formatting. Shorten repetitive sections, standardize tables and figures, and fix minor English errors.

Lines 1–3:

“The Association between Novel Anthropometric Indices and Cognitive Function in Iranian Older Adults…”

Comment: The title accurately reflects the topic but is lengthy. A concise version such as “Novel Anthropometric Indices and Cognitive Function among Iranian Older Adults” would improve clarity and alignment with PLOS ONE’s concise title policy.

Lines 4–22:

Comment: Multiple affiliations are clearly presented; however, ensure consistent formatting of institutional units (e.g., use “Tehran University of Medical Sciences, Tehran, Iran” uniformly).

Abstract (Lines 36–64)

Lines 38–41:

The background succinctly introduces TyG, VAI, and LAP, but it would benefit from a stronger rationale explaining why these indices might predict cognitive impairment (e.g., connection to insulin resistance mechanisms).

Lines 43–49 – Methods:

Specify that logistic regression was sex-stratified a priori to justify separate male/female analyses. Also indicate that confounders were selected via backward stepwise regression.

Lines 50–56 – Results:

Results are clearly stated but slightly overloaded with numeric detail. Report only the key significant results with directionality; move others to the Results section. Include the total cognitive impairment prevalence (63.9%) only once.

Lines 57–60 – Conclusion:

Avoid causal language (“highlight the interplay”) and rephrase to reflect association: “These results suggest complex, sex-specific associations between metabolic indices and cognition.”

**Add the study design (“cross-sectional”) explicitly in the first sentence of the Abstract.

Introduction (Lines 65–99)

Lines 69–71:

Strong epidemiological framing. Add a recent Iranian study (within the last 3 years) to justify local context.

Lines 75–92:

Excellent description of insulin resistance and novel indices. However, several sentences could be condensed to reduce redundancy (e.g., references 11–17 repeat the mechanistic explanation).

Lines 93–97:

The final paragraph should explicitly define the gap in prior literature—few studies examining TyG, VAI, and LAP together in non-diabetic older adults—and restate the novelty more assertively.

Study Design (101–118):

•Clarify timeline: The manuscript says “phase 2 of baseline ”measurements”—this phrasing is confusing. Specify if this is the second data collection wave.

•Mention ethical approval and consent here briefly, then expand later.

•State explicitly that the analysis was conducted in 2025 and that no identifiable data were accessed.

Data Collection (119–133):

•Sentence structure unclear (“Data collection included age, sex, income, and level of education, which were gathered…”). Revise for grammatical accuracy.

•Define how income was categorized.

•The PCA-based socio-economic index is mentioned but without variance explained; report the percentage of variance accounted for by the first principal component.

Lifestyle and Physical Activity (128–133):

•Include the name of the validated questionnaire used (e.g., Global Physical Activity Questionnaire, IPAQ, etc.).

•The MET categories differ from WHO thresholds—justify the classification system or cite a local validation.

Cognitive Tests (134–152):

•Excellent that three instruments were used. However, the combination rule (“impairment in one or more tests”) could inflate prevalence; justify this decision.

•Report Cronbach’s α, or internal reliability, for each translated test in the current sample.

•Replace “demantia” (typo, line 140) with “dementia.”

Physical and Laboratory Measures (153–172):

•Add the year of calibration or QA/QC procedure for lab assays.

•Minor typographical error: “measuremnts” → “measurements.”

•Define units consistently (e.g., mmol/L or mg/dL, but not both in formulae).

Novel Anthropometric Indices (173–184):

•Formulas are accurate. For transparency, provide conversion factors between mg/dL and mmol/L where applicable.

•Clarify whether lipid and glucose measures were log-transformed before inclusion in models.

Statistical Analysis (185–197):

•Excellent inclusion of regression modeling. However:

oAdd model diagnostics (e.g., Hosmer–Lemeshow test, multicollinearity check).

oReport how missing data were handled (listwise deletion, imputation?).

oAvoid “backward stepwise”; instead, use theoretical or evidence-based selection to avoid model overfitting.

oState the statistical power or effect size detectable given N=2,426.

Table 1 (225–226):

•Formatting issue: inconsistent column spacing and misaligned p-values.

•Many p-values are reported without correction for multiple testing; consider FDR adjustment.

•Provide units in table headers (e.g., “BMI (kg/m²)”).

•The table is overcrowded; consider splitting demographic and biochemical variables.

Lines 213–219:

The text correctly describes significant group differences but overuses repetition. Condense to highlight major findings (e.g., higher prevalence of impairment in females, p < 0.001).

Lines 228–246:

•Report model goodness-of-fit or pseudo-R² for logistic regressions.

•Use consistent precision (two decimals for OR, two for CI limits).

•Clarify whether sex-stratified analyses were exploratory or planned.

Tables 2–4 (Lines 247–252):

•Tables are detailed but overly complex; emphasize key associations in text rather than reproducing all.

•Clearly label adjusted covariates in each legend.

•Highlight significant findings (e.g., p < 0.05) using bold or footnotes rather than asterisks scattered inconsistently.

•Verify that confidence intervals match p-values; some do not align (e.g., p=0.4 but CI excludes 1).

Discussion (Lines 253–313)

Lines 253–258 – Summary:

Concise and appropriate; however, avoid restating numeric results. Instead, emphasize direction and magnitude.

Lines 259–278 – TyG:

Well contextualized. Consider including a physiological rationale for lack of association (e.g., TyG reflects peripheral IR, not central nervous system IR). Current explanation is accurate but could be tightened.

Lines 279–292 – VAI:

Balanced, but the section mixes discussion of literature with results. Suggest splitting into “Comparison with prior studies” and “Potential mechanisms.”

Add a hypothesis why associations differ by sex (e.g., estrogenic modulation, body fat distribution).

Lines 293–299 – LAP:

Good synthesis, but a brief comment on potential multicollinearity between indices (VAI includes BMI and TG, which also feed into LAP) would enhance rigor.

Lines 300–305 – Integrative interpretation:

Excellent conceptual framing; could be improved by mentioning inflammation, oxidative stress, or adipokine dysregulation as shared mechanisms.

Lines 306–313 – Limitations:

Comprehensive. You may also add: “Residual confounding from unmeasured variables such as diet quality and APOE genotype cannot be excluded.”

Conclusion (Lines 314–320)

Sound and cautious, but slightly repetitive. Revise to two sentences:

“In this large cross-sectional study of Iranian older adults, novel anthropometric indices (TyG, LAP, VAI) were not consistently associated with cognitive impairment. Observed sex- and test-specific differences suggest the need for longitudinal studies to clarify causal pathways.”

References

•Ensure consistency in reference formatting.

•Add 3–4 recent studies (2023–2025) to strengthen currency.

https://doi.org/10.1007/s40200-024-01404-8

https://doi.org/10.1002/ncp.11273

https://doi.org/10.1016/j.puhe.2025.01.040

•Verify numbering sequence.

Reviewer #3: I have reviewed the manuscript entitled “The Association between Novel Anthropometric Indices and Cognitive Function in Iranian Older Adults: Bushehr Elderly Health (BEH) Program.” This study presents a cross-sectional analysis utilizing data collected from 2,426 participants aged 60 and older as part of the Bushehr Elderly Health program. Cognitive function was assessed through the Functional Assessment Staging Test (FAST), Mini-Cog, and Category Fluency Test (CFT) questionnaires. A key finding of this research is the gender-specific association between the Visceral Adiposity Index (VAI) and the results from the FAST, particularly in male participants.

The FAST primarily focuses on functional abilities rather than cognitive testing, evaluating various aspects of functioning from independence in basic activities of daily living to more advanced stages of dementia. However, it may not effectively detect mild cognitive impairment in the early stages of the disease process because of its emphasis on functional decline. Additionally, scoring can be subjective, reliant on the observer’s interpretation of an individual’s functioning, and it does not directly measure cognitive function, which may restrict its usefulness in diagnosing specific types of cognitive impairment. The Mini-Cog is a brief cognitive screening tool that assesses recall ability. It is quick and straightforward to administer and shows particular effectiveness in detecting dementia among older adults. The test requires minimal language skills, but its results may be influenced by cultural familiarity with time-telling devices and can lead to false negatives, particularly in the early stages of cognitive decline. The CFT evaluates verbal fluency and is often correlated with more comprehensive cognitive assessments, which enhances its utility in identifying cognitive impairment. However, the results can be affected by factors such as educational level, language proficiency, and cultural background, and performance may fluctuate based on the individual’s current state (e.g., fatigue or anxiety). Given the measurement limitations related to the primary outcomes and the inherent confounding factors associated with the cross-sectional design of this study, interpreting the results presents challenges. A cohort study design may be more appropriate for investigating the research topic at hand. I commend the authors for their significant efforts in conducting this study; however, several concerns and requests for clarification arise. Below are my observations and suggestions for further elaboration:

1. It is essential to provide a separate table that details the all basic characteristics and variables of participants, comparing men and women without subgrouping while reporting the P-values.

2. A thorough discussion of the advantages and limitations of the FAST, Mini-Cog, and CFT, as above mentioned, should be integrated into the discussion section. Additionally, consider addressing the potential influence of confounding factors, such as educational status, functional abilities, and physical health, which may differ by gender.

3. Although this study did not measure other non-insulin based indices of insulin resistance (IR) such as HOMA-IR and TG/HDL, explaining the results mainly through IR mechanisms, which are a vital component of the discussion section, indicates a strong need to revise the discussion section to focus more on the factual gender-specific differences in participant performance in tests that evaluate cognitive impairment.

4. I recommend including a recruitment flow chart that depicts the percentage of excluded participants at each stage, especially the proportion of missing laboratory values.

5. To enhance clarity, please rephrase “co-association” in line 304.

6. PLOS authors have the option to publish the peer review history of their article (what does this mean?). If published, this will include your full peer review and any attached files.). If published, this will include your full peer review and any attached files.

.

Reviewer #1: No

Reviewer #2: No

Reviewer #3: **Yes:** Shahin AbbaszadehShahin Abbaszadeh

---

## [Author Response · Author response to Decision Letter 1]

15 Dec 2025

Dear editorial manager and Reviewers:

Thank you for providing comments on our manuscript entitled “Novel Anthropometric Indices and Cognitive Function among Iranian Older Adults: Bushehr Elderly Health (BEH). Those comments are all valuable and very helpful for revising and improving our manuscript, as well as providing important guidance to our research.

PLOS ONE style requirements: We have fully reformatted the manuscript (main text, title page, figures, and tables) according to the official PLOS ONE templates provided in the links.

Data Availability:

The data used in this study are from the Bushehr Elderly Health (BEH) program, a population-based cohort study owned and governed by the Ministry of Health and Medical Education of the Islamic Republic of Iran through the Persian Cohort and Biobank regulations. Due to ethical and legal restrictions on sharing potentially identifiable human data and national regulations, the raw dataset cannot be made publicly available. However, the de-identified minimal dataset underlying the results of this study is available upon reasonable request to the corresponding author.

This statement is also added to the manuscript.

Regards

Authors

Reviewer Comments to the Author

Reviewer #1:

1. Some sentences are long and could be broken into shorter sentences for readability.

Response: Thank you. We have carefully checked the entire manuscript and edited long sentences into shorter ones where possible to improve readability. Please let us know if you identify any remaining areas that need further adjustment. (Whole manuscript)

2. References to prevalence in Iran could include more recent statistics if available.

Response: We have added more recent statistics on prevalence of cognitive impairment in Iran, including references 4–6, to ensure the data are up to date.

3. Consider emphasizing why these indices are practical alternatives to HOMA-IR (cost and accessibility) a bit earlier to strengthen the rationale.

Response: We have emphasized earlier that these indices are practical alternatives to HOMA-IR due to their lower cost and greater accessibility. (Lines 87-89)

4. Consider summarizing borderline associations separately from statistically significant results to improve clarity.

Response: Thanks. We decided to remove some of non-significant associations from the text.

(Lines252-255)

5. Some sentences are repetitive (e.g., noting no association for TyG and LAP in multiple places). Consolidating these points could improve readability.

Response: Thank you for the comment. We have carefully reviewed the manuscript and edited the repetitive sentences in the Discussion and Conclusion sections to improve readability. (Lines273-275, Lines302-310, Lines 371-374).

6. Clarify whether quartile-based analyses were adjusted for multiple comparisons.

Response: Thank you for the comment. To examine this relationship, we conducted quantitative and qualitative analysis, following the precedent of similar studies (https://doi.org/10.1186/s12944-023-01959-0).

Reviewer #2:

•The study design is cross-sectional, yet several statements imply causality (e.g., “interplay between metabolic indices and cognitive function”). These should be rephrased to reflect association, not cause–effect.

Response: We have reviewed the manuscript and revised the relevant statements to emphasize association rather than causation or “interplay,” ensuring alignment with the cross-sectional design. If any phrasing still appears causal, please let us know and we will adjust it further. (Whole manuscript)

•Some methodological details are insufficient: the phrase “phase 2 of baseline measurements” is ambiguous; it should clearly indicate whether this was a second wave of data collection or a subsample of a longitudinal cohort.

Response: Thank you for the comment. We have clarified the timeline by explicitly stating that Stage 2 represents the second wave of baseline data collection. (Lines 104-105)

•Definitions of cognitive impairment (based on impairment in ≥1 of 3 tests) may overestimate prevalence; justification or sensitivity analysis using stricter criteria is needed.

Response: Thank you for the comment. To maximize sensitivity for case inclusion, the study was informed by the established definitions and consensus criteria of all three assessment tools.

•Despite reporting significant associations, the biological and clinical relevance of observed odds ratios (close to 1.0) is minimal and should be interpreted cautiously.

Response: Thank you for your valuable comment. We have added this point to our limitation section, and we also reviewed the entire manuscript to ensure that any statistically significant findings with minimal clinical relevance were interpreted cautiously and non-significant ones were not reported in the text. (Lines 321-326, Lines 65-67)

•The manuscript states that covariates were selected using backward stepwise regression, which can lead to unstable models and overfitting. Covariate inclusion should be theory-driven or justified by prior evidence.

Response: Thank you and respect: In this analysis, variables identified in previous studies as potential confounders for cognitive impairment were first extracted. Subsequently, all of these variables were included in the logistic regression model. Variables with a p-value greater than 0.2 were then removed through a backward elimination process, allowing only the significant variables to remain in the final model. In the next step, the adjusted logistic regression model was developed based on these key variables. (Lines 214-218).

•The paper does not describe how missing data were handled. State whether listwise deletion, imputation, or other methods were used.

Response: Thank you for the comment. Edited. The listwise deletion method was used to handle missing data. (Lines 124-129)

•No mention of multicollinearity testing, goodness-of-fit (e.g., Hosmer–Lemeshow test), or sensitivity analyses is provided.

Response: Thank you for the comment. Edited. (Tables 3,4,5)

•Effect sizes and confidence intervals are reported but should be discussed in terms of practical importance.

Response: We appreciate the reviewer's suggestion. We have revised the discussion section to interpret the effect sizes in terms of their modest practical importance, emphasizing the need for additional risk factors and further research to assess clinical relevance. (Lines 321-326, Lines 365-367)

•The decision to stratify by sex is reasonable, but the rationale for doing so should be stated explicitly and, ideally, supported by a statistical test for interaction.

Response: Thank you and respect. Given the significant interactions observed between gender and cognitive impairment with each of the lipid profile components, and considering the potential role of sex as an effect moderator in the association between lipid profile and psychological/cognitive impairment, all analyses were stratified by sex.

•The manuscript is understandable, but several sentences are long, redundant, or grammatically inconsistent.

Response: Thank you. We have carefully checked the entire manuscript and edited long sentences into shorter ones where possible to improve readability. Please let us know if you identify any remaining areas that need further adjustment.

•Minor typographical errors occur (“demantia,” “measuremnts”).

Response: Thank you. Edited. (Line152-Line 175).

•Paragraphs in the Discussion are overly descriptive; the narrative should be condensed and more analytical.

Response: Thank you for the valuable feedback. We have condensed overlay data, and added deeper interpretation of the findings, comparisons with previous studies. (Lines 276-286,

Lines 352-361).

•Ensure consistent formatting of tables, units (mmol/L vs mg/dL), and reference style.

A professional English-language edit is recommended before resubmission.

Response: We have thoroughly reviewed and edited the manuscript. All values are now uniformly reported in their standard format, with conversion factors added to the Methods section. Table 1 presents demographic characteristics, Table 2 reports all measures in their standard units, and Tables 3–5 present odds ratios with 95% confidence intervals. (Lines 186-203,Tables1-5)

1.Clarify the study design and population. Clearly describe what “phase 2 of baseline” refers to and specify inclusion/exclusion criteria.

Response: Thank you for the comment. We have clarified the timeline by explicitly stating that Stage 2 represents the second wave of baseline data collection. (Lines 104-113)

2.Refine the definition of cognitive impairment. Provide justification for your threshold (impairment in ≥1 test) and test an alternative stricter definition in sensitivity analysis.

Response: Thank you for your feedback. First, the definition of cognitive impairment was applied separately to each diagnostic tool. Then, to maximize sensitivity and ensure inclusion of all potential cases of cognitive impairment, a consensus criterion was used that spanned all three assessment tools.

3.Re-examine model specification. Replace stepwise regression with theory-driven models or demonstrate that results are robust to different covariate sets.

Response: Thank you for the comment: In this analysis, variables identified in previous studies as potential confounders for cognitive impairment were first extracted. Subsequently, all of these variables were included in the logistic regression model. Variables with a p-value greater than 0.2 were then removed through a backward elimination process, allowing only the significant variables to remain in the final model. In the next step, the adjusted logistic regression model was developed based on these key variables. (Lines 214-218)

4.Address data handling. Explain how missing values were treated and whether multicollinearity was assessed.

Response: Thank you for the comment. Edited. The listwise deletion method was used to handle missing data. (Lines 124-129)

5.Add regression diagnostics. Include measures such as pseudo-R² and goodness-of-fit statistics.

Response: Thank you for the comment. Edited. (Tables 3,4,5)

6.Report and interpret effect sizes more cautiously. ORs near 1.0 may be statistically significant but clinically negligible.

Response: Thank you for your valuable comment. We have added this point to our limitation section, and we also reviewed the entire manuscript to ensure that any statistically significant findings with minimal clinical relevance were interpreted cautiously and non-significant ones were not reported in the text. (Lines 321-326, Lines 367-367)

7.Revise discussion and conclusion. Avoid causal language; discuss mechanisms and confounders (sex hormones, fat distribution, inflammation).

Response: We have revised the discussion and conclusion to avoid causal language while adding more detail on potential mechanisms and confounders. (Lines 276-286,

Lines 347-356)

8.Ensure open data compliance. Deposit anonymized dataset in a public repository and update the Data Availability Statement.

Response: We have updated the Data Availability Statement to indicate that the anonymized dataset has been deposited in a public repository for open data compliance. (Lines 391-396)

9.Language and formatting. Shorten repetitive sections, standardize tables and figures, and fix minor English errors.

Response: Thank you for the comment. We have thoroughly reviewed and edited the manuscript. All values are now uniformly reported in their standard format, with conversion factors added to the Methods section. (Whole manuscript)

Lines1–3:

“The Association between Novel Anthropometric Indices and Cognitive Function in Iranian Older Adults…”

Comment: The title accurately reflects the topic but is lengthy. A concise version such as “Novel Anthropometric Indices and Cognitive Function among Iranian Older Adults” would improve clarity and alignment with PLOS ONE’s concise title policy.

Response: We really appreciate your comment. The title is edited. (Lines 1-3)

Lines4–22:

Comment: Multiple affiliations are clearly presented; however, ensure consistent formatting of institutional units (e.g., use “Tehran University of Medical Sciences, Tehran, Iran” uniformly).

Response: Thanks. Rechecked. (Lines 4-22)

Abstract

Lines38–41:

The background succinctly introduces TyG, VAI, and LAP, but it would benefit from a stronger rationale explaining why these indices might predict cognitive impairment (e.g., connection to insulin resistance mechanisms).

Response: Thank you for your comment. Added. (Lines 40-43)

Lines43–49–Methods:

Specify that logistic regression was sex-stratified a priori to justify separate male/female analyses. Also indicate that confounders were selected via backward stepwise regression.

Response: Thanks. Edited. (Lines 50-53)

Lines 50–56 – Results:

Results are clearly stated but slightly overloaded with numeric detail. Report only the key significant results with directionality; move others to the Results section. Include the total cognitive impairment prevalence (63.9%) only once.

Response: Some of the less essential data has been removed. (Lines 55-59)

Lines57–60–Conclusion:

Avoid causal language (“highlight the interplay”) and rephrase to reflect association: “These results suggest complex, sex-specific associations between metabolic indices and cognition.”

Response: Thank you for your precision. Edited. (Line 62)

**Add the study design (“cross-sectional”) explicitly in the first sentence of the Abstract.

Response: Added. (Line38)

Introduction

Lines69–71:

Strong epidemiological framing. Add a recent Iranian study (within the last 3 years) to justify local context.

Response: Thank you for the suggestion. The sentence has been edited and a recent reference has been added. (Lines73-74)

Lines75–92:

Excellent description of insulin resistance and novel indices. However, several sentences could be condensed to reduce redundancy (e.g., references 11–17 repeat the mechanistic explanation).

Response: Thank you for the suggestion. Some of the sentences have been condensed. (Lines 85-87, Lines 90-92)

Lines93–97:

The final paragraph should explicitly define the gap in prior literature—few studies examining TyG, VAI, and LAP together in non-diabetic older adults—and restate the novelty more assertively.

Response: Thank you for the comment. The paragraph has been revised to clearly define the literature gap and emphasize the novelty of our study. (Lines 95-101).

StudyDesign(101–118):

•Clarify timeline: The manuscript says “phase 2 of baseline ”measurements”—this phrasing is confusing. Specify if this is the second data collection wave.

Response: Thank you for the comment. We have clarified the timeline by explicitly stating that Stage 2 represents the second wave of baseline data collection. (Lines 104-105)

•Mention ethical approval and consent here briefly, then expand later.

Response: Thank you for mentioning that. Added. (Lines 118-120)

•State explicitly that the analysis was conducted in 2025 and that no identifiable data were accessed.

Response: Thanks. We have explicitly stated that the analysis was conducted in 2025 and that only de-identified data were used. (Lines 122-124)

DataCollection(119–133):

•Sentence structure unclear (“Data collection included age, sex, income, and level of education, which were gathered…”). Revise for grammatical accuracy.

Response: We really appreciate your precision. Edited. (Lines 131-132)

•Define how income was categorized.

Response: Income was collected through direct survey questions and was not used as a standalone categorical variable; instead, it was incorporated as a component of the socioeconomic status index derived using principal component analysis (PCA). (Lines 131-132)

•The PCA-based socio-economic index is mentioned but without variance explained; report the percentage of variance accounted for by the first principal component.

Response: We have now reported the variance explained by the first principal component; it accounted for 52.0% of the total variance. (Lines 134-137)

Lifestyle and Physical Activity (128–133):

•Include the name of the validated questionnaire used (e.g

---

## [Decision Letter · Decision Letter 1]

6 Feb 2026

Novel Anthropometric Indices and Cognitive Function among Iranian Older Adults: Bushehr Elderly Health (BEH) Program

PONE-D-25-56488R1

Mahnaz Pejman Sani

Dear Dr. Mahnaz pejman Sani,

We’re pleased to inform you that your manuscript has been judged scientifically suitable for publication and will be formally accepted for publication once it meets all outstanding technical requirements.

An invoice will be generated when your article is formally accepted. Please note, if your institution has a publishing partnership with PLOS and your article meets the relevant criteria, all or part of your publication costs will be covered. Please make sure your user information is up-to-date by logging into Editorial Manager at Editorial Manager® and clicking the ‘Update My Information' link at the top of the page. For questions related to billing, please contact  and clicking the ‘Update My Information' link at the top of the page. For questions related to billing, please contact billing support..

Kind regards,

Marwan Salih Al-Nimer, MD, PhD

Academic Editor

PLOS One

Additional Editor Comments (optional):

Reviewers' comments:

Reviewer's Responses to Questions

**Comments to the Author**

1. If the authors have adequately addressed your comments raised in a previous round of review and you feel that this manuscript is now acceptable for publication, you may indicate that here to bypass the “Comments to the Author” section, enter your conflict of interest statement in the “Confidential to Editor” section, and submit your "Accept" recommendation.

Reviewer #3: All comments have been addressed

2. Is the manuscript technically sound, and do the data support the conclusions?

Reviewer #3: Yes

3. Has the statistical analysis been performed appropriately and rigorously? 

Reviewer #3: Yes

4. Have the authors made all data underlying the findings in their manuscript fully available?

Reviewer #3: No

5. Is the manuscript presented in an intelligible fashion and written in standard English?

Reviewer #3: Yes

6. Review Comments to the Author

Reviewer #3: (No Response)

7. PLOS authors have the option to publish the peer review history of their article (what does this mean?). If published, this will include your full peer review and any attached files.). If published, this will include your full peer review and any attached files.

.

Reviewer #3: **Yes:** Shahin AbbaszadehShahin Abbaszadeh

---

## [Editor Report · Acceptance letter]

PONE-D-25-56488R1

PLOS One

Dear Dr. Pejman Sani,

I'm pleased to inform you that your manuscript has been deemed suitable for publication in PLOS One. Congratulations! Your manuscript is now being handed over to our production team.

Kind regards,

on behalf of

Professor Marwan Salih Al-Nimer

Academic Editor

PLOS One